# OpenReview forum: "ROBBIE: Robust Bias Evaluation of Large Generative Language Models"
_EMNLP/2023/Conference — EMNLP 2023 Main_

### Official Review · Reviewer_4RU4 · 2023-08-04

**Soundness:** 4

**Excitement:**

4: Strong: This paper deepens the understanding of some phenomenon or lowers the barriers to an existing research direction.

**Paper Topic And Main Contributions:**

The authors propose an adversarial benchmark suite, ROBBIE, that analyzes the performance of LLMs across 12 demographic axes and 6 datasets. It also compares different mitigation techniques and shows their effectiveness.

**Questions For The Authors:**

Are the results shown in Table 4 on "intersectional identities" of demographic labels or on the co-occurrence of demographic labels? For example, the difference between the sentences -- "It’s like the homosexuals showing up at the Christian bakery.” vs "It’s like the Asian homosexuals showing up at the bakery" -- The latter would be an intersectional identity but the former is a sentence where two demographic attributes are present but they are independent of each other. I'm curious to know if the authors conducted an error analysis to see how many sentences actually had intersectional identities and how many of the sentences in the benchmark just have two or more demographic attributes co-occurring in the sentence.

**Reasons To Accept:**

+ Propose a ROBBIE benchmark suite that analyzes performance of LLMs across 12 demographic axes and 6 datasets.
+ Extend prompt based metrics to intersection of demographic groups.
+ Provide insights into the pre-training of the model corpora and how that might relate to model bias.

**Reasons To Reject:**

- Usage of the term "bias". I'd recommend stating the kinds of harm the authors are trying to measure instead of using an all-encompassing term "bias". The authors provide a definition of "bias" in the introduction, but that could be improved by explicitly stating the harms the authors measure.
- Details about dataset creation should move to the main paper from the appendix

**Reproducibility:**

3: Could reproduce the results with some difficulty. The settings of parameters are underspecified or subjectively determined; the training/evaluation data are not widely available.

**Reviewer Confidence:**

3: Pretty sure, but there's a chance I missed something. Although I have a good feel for this area in general, I did not carefully check the paper's details, e.g., the math, experimental design, or novelty.

---

> ### Author Rebuttal · Authors · 2023-08-29
>
> Thank you for your positive review! We appreciate that you found our paper exciting with strong soundness, providing insights into performance of LLMs across 12 demographic axes and 6 datasets, and how pre-training data relates to model bias.
>
> Below please find our honest effort to address your concerns.
>
> > Usage of the term "bias". I'd recommend stating the kinds of harm the authors are trying to measure instead of using an all-encompassing term "bias". The authors provide a definition of "bias" in the introduction, but that could be improved by explicitly stating the harms the authors measure.
>
> Thank you for your valuable feedback. We will make necessary amendments to our manuscript to enhance the definition of "bias" by explicitly outlining the specific forms of harm that our study aims to quantify, as you have suggested. In our current draft, the definition of bias is presented in section 3.1 at line 263. In essence, we characterize bias as the proportion of subgroups for which the frequency of toxicity and negative regard falls outside an acceptable threshold. This definition is rooted in the principle of demographic parity, serving as a benchmark for equality and fairness, as previously applied in the context of fairness assessment within Natural Language Processing (NLP).
>
> > Details about dataset creation should move to the main paper from the appendix.
>
> We thank the reviewer for the great suggestion. We agree that the adding details about the dataset creation in the main paper will improve the readability of the paper. As stipulated by the submission guidelines, we are currently prohibited from making any revisions to the paper prior to receiving the acceptance or rejection notification. However, if our paper is accepted, we are committed to implementing the proposed changes. Specifically, we will transfer the comprehensive information about the dataset creation, detailed in section B.1.2, along with Figure 2, to the main body of the paper.
>
> > Are the results shown in Table 4 on "intersectional identities" of demographic labels or on the co-occurrence of demographic labels? For example, the difference between the sentences -- "It’s like the homosexuals showing up at the Christian bakery.” vs "It’s like the Asian homosexuals showing up at the bakery" -- The latter would be an intersectional identity but the former is a sentence where two demographic attributes are present but they are independent of each other. I'm curious to know if the authors conducted an error analysis to see how many sentences actually had intersectional identities and how many of the sentences in the benchmark just have two or more demographic attributes co-occurring in the sentence.
>
> In response to your **question**, we thank you for your thought-provoking query regarding the distinction between "intersectional identities" and the co-occurrence of demographic labels in our study. While we didn't conduct a specific error analysis to differentiate these aspects, we did implement a filter during data preprocessing to exclude instances where multiple traits appeared on the same demographic axis. This aimed to isolate singular identity cases, yet we acknowledge the potential limitations of this approach in capturing true intersectionality. Due to time and budget constraints, a more granular error analysis wasn't feasible in this study. We do recognize the importance of addressing this limitation and enhancing our methodology. In our future work, we plan to refine our dataset by assigning demographic labels per entity, enabling a more accurate identification of intersectional instances. Your feedback has been invaluable in highlighting this aspect, and we are committed to transparently addressing it in our paper. Your engagement underscores the ongoing evolution of our research, and we deeply appreciate your contribution.
>
> > Could reproduce the results with some difficulty. The settings of parameters are underspecified or subjectively determined; the training/evaluation data are not widely available.
>
> Regarding you concern about **reproducibility**, we would like to iterate that we are committed to open sourcing the proposed two new datasets (AdvPromptSet and HolisticBiasR) as well as the evaluation pipeline and bias and toxicity mitigation toolkit for robust measurement across different metrics, fostering reproducibility and benefiting the broader research community. This initiative reinforces the reliability of our paper's findings. The other four metrics that are compared in the paper are public research benchmarks. We have also detailed our decoding parameter settings in B.2 and settings of different bias/toxicity mitigation techniques in B.3 to enable the reproducibility of our paper.
>
> Thank you again for your thoughtful review and insightful questions, we hope we were able to answer your questions and address your concerns. Please don’t hesitate to let us know if there are any additional questions or concerns.

---

### Official Review · Reviewer_n5Eu · 2023-08-04

**Soundness:** 4

**Excitement:**

3: Ambivalent: It has merits (e.g., it reports state-of-the-art results, the idea is nice), but there are key weaknesses (e.g., it describes incremental work), and it can significantly benefit from another round of revision. However, I won't object to accepting it if my co-reviewers champion it.

**Paper Topic And Main Contributions:**

This paper brings a study of toxicity and undesired bias in recent generative language models. It proposes a wide variety of demographic axes and identifies the discriminative features for them in the previous datasets. Using the proposed methodology, it evaluates five recent language models, together with three bias mitigation techniques applicable to the inference. Finally, it underlines some trends, such as that the larger models are more reliant on toxic features than the smaller ones.


**Questions For The Authors:**

A: In Section 3.2, you also confronted the human evaluation results, but I could not find any quantified reports regarding this. Are the details of the reported human evaluation available somewhere?

**Reasons To Accept:**

This is a very wide and thorough study of many covariates, bias features and their combinations packed in a single paper.
* The flow of the paper is easy to follow, with well-commented motivation in Introduction and attached to all major experiments
* Most important experimental settings (such as models selection, toxicity axes, and used metrics) are justified in the text
* The volume of experiments performed and processed within this work is, in my opinion, outstanding
* Whenever appropriate, the authors also include the discussion of the limitations directly in the main text (e.g. L451-455). This helps maintain a realistic and honest framing of the paper's scope and implications, which I really appreciate.
* I believe that the ability to perform evaluations over mixtures of the groups (such as Race x Gender in Table 4) that the proposed AdvPromptSet dataset enables is interesting and important for the field.


**Reasons To Reject:**

* Many results presented in the paper do not contain experimental details and results directly in the main text, but point the reader to the appendices. The descriptions of the experiments remain focused to the main objectives, but this makes it difficult to closely follow what is going on. Conceptually, I would prefer to prioritize some experiments that authors consider more important and describe them in detail in-place, even for the price of narrowing the broadness of results presented in the main text.
* Authors clearly state and rationalize (L214-218) why they do not include the evaluation of instruction-tuned models among evaluation. However, I am not perfectly convinced by the rationalization, as the instruction tuning might change the toxicity inclinations of the model, and instruction tuning, as well as RLHF, might be necessary for the use of evaluated foundational models for many end applications.
* I am not convinced of the assumption that more complex models require more complex analyses, which also appears in the abstract (L001-004), which is pursued from a case study of BlenderBot (L426-429). Unless I missed some other support, I suggest restraining from presenting this as a fact.
* From the broadness of the experiments, I would expect the emergence of clearer conclusions about the covariates of LMs' toxicity. Apart from confirming certain dependencies on the model size, I could not identify any clear takeaways in the Conclusion section.
* I can not assess the practical value of this paper to followup work given that the source code of the analyses is not referenced, nor is it available in supplementary materials. Regretfully, the sole promise of publishing it after the peer review does not allow for the evaluation of its practical usability.


**Reproducibility:**

1: Could not reproduce the results here no matter how hard they tried.

**Reviewer Confidence:**

3: Pretty sure, but there's a chance I missed something. Although I have a good feel for this area in general, I did not carefully check the paper's details, e.g., the math, experimental design, or novelty.

---

> ### Author Rebuttal · Authors · 2023-08-29
>
> Thank you for your careful and well-organized review! We are thrilled that you found our study to be comprehensive and well-structured, and are delighted that you share our enthusiasm for the evaluation of intersection of demographics enabled by the AdvPromptSet dataset.
>
> Below please find our honest effort to address your concerns.
>
> > Many results presented in the paper do not contain experimental details and results directly in the main text, but point the reader to the appendices. The descriptions of the experiments remain focused to the main objectives, but this makes it difficult to closely follow what is going on. Conceptually, I would prefer to prioritize some experiments that authors consider more important and describe them in detail in-place, even for the price of narrowing the broadness of results presented in the main text.
>
> We would like to stress that the focus of our paper is two folds: 1) **Benchmarking:** a comparison of 6 different prompt-based bias and toxicity metrics across 12 demographic axes and 5 families of generative LLMs. 2) **Mitigation:** the comparison of those benchmarks gives us insights about the bias and toxicity of the compared models. Therefore, we conduct a comprehensive study of how well 3 bias and toxicity mitigation techniques perform across our suite of measurements.
> In order to achieve our goal, we have included a broad set of the major results in the main text. We understand that there may be some references in the tables or figures that point readers to the appendix. Our original intention was to provide a comprehensive discussion given the broadness of results. If the paper is accepted, we will reorganize the paper for a clearer presentation. To avoid distraction and digression of the main results, we plan to reduce the number of references to the appendix in our tables and figures. For example, in Figure 1, we will remove the sentence of “Table 9 gives the full table of values.” because the figure itself is already self-contained to toxicity and negative regard of different model families on various benchmarks.
>
> > Authors clearly state and rationalize (L214-218) why they do not include the evaluation of instruction-tuned models among evaluation. However, I am not perfectly convinced by the rationalization, as the instruction tuning might change the toxicity inclinations of the model, and instruction tuning, as well as RLHF, might be necessary for the use of evaluated foundational models for many end applications.
>
> There are a lot more open-sourced models that do not have RLHF yet, and at the time of our work/paper release, many RLHF models are not open-sourced. In addition, many RLHF’d or instruction fine-tuned models are trailed towards taking instructions, and for some prompt completion tasks we have, we would need to twist the prompts to meet/trigger the changed distribution. Therefore we believe not evaluating on RLHF/instruction fine-tuned model will not affect our key takeaways, which is that we need a more holistic approach to evaluate bias, rather than single metrics. This work will definitely open more research opportunities to evaluate on the RLHF/instruction fine-tuned models, now that there are more of such models that are open-sourced.
> We acknowledge your concern regarding our exclusion of instruction-tuned models from evaluation, despite our stated rationale (L214-218). While we recognize the importance of evaluating RLHF models, we focused our study on foundational models without such tuning due to several reasons. At the time of our research and paper release, many RLHF models were not open-sourced, limiting our ability to include them in our evaluation. Additionally, RLHF and instruction fine-tuned models are often tailored for specific instruction-following tasks, and adapting prompts to accommodate their changed distributions could introduce complexities into our evaluation framework, potentially altering the context of our study.
> However, we firmly believe that our central message, advocating for a holistic approach to bias evaluation, stands independently of RLHF model evaluations. Our work aims to underscore the need for comprehensive assessment beyond singular metrics. It's encouraging to note your view that the emergence of open-source RLHF models offers new avenues for research. We share your sentiment that the evolving landscape presents opportunities for future exploration in this domain.
>
> > I am not convinced of the assumption that more complex models require more complex analyses, which also appears in the abstract (L001-004), which is pursued from a case study of BlenderBot (L426-429). Unless I missed some other support, I suggest restraining from presenting this as a fact.
>
> We sincerely appreciate your feedback and are committed to revising our abstract to ensure the clear communication of our intended message.  We also extend our apologies for any confusion that may have arisen. Our original intention was not to suggest that more complex models require more complex analysis. Instead, our objective is to underscore the gaps present in each benchmark and highlight the enhanced insights attainable through the combined utilization of these benchmarks. This approach enables us to uncover a more comprehensive array of risks stemming from Language Models (LLMs) from a more holistic perspective. This also underscores the imperative for improved tools designed to gauge fairness and bias across all LLMs. As the prevalence of LLMs grows, our focus intensifies on the development of more refined tools for measuring and mitigating toxicity and bias concerns.
>
> > From the broadness of the experiments, I would expect the emergence of clearer conclusions about the covariates of LMs' toxicity. Apart from confirming certain dependencies on the model size, I could not identify any clear takeaways in the Conclusion section.
>
> Thank you for pointing out the need for clarity with respect to our key takeaways and the need to restructure our conclusion sections to reflect that. We would like to list the main insights from our work which will be reflected in the revision of the Conclusion section:
> - **Impact of Model Size on Bias and Toxicity:** Toxicity and negative regard often increases as a function of model size, but not always. Our findings align with previous research, and highlights the importance of conducting robust bias analysis on modern LLMs.
> - **Consistent bias towards specific Demographics:** Our analysis revealed a consistent negativity towards some demographics such as Hispanic or Latino across all LLMs. Considering the growing integration of LLMs into our daily lives, this bias could have significant implications for these minority groups.
> - **Differential Effects of Benchmark Datasets:** Each prompt dataset led to varying levels of BiasScore, toxicity and negative regard in the generated content of LLMs. The substantial bias indicated by our BiasScore results across all LLMs highlights the necessity of employing diverse benchmark datasets to uncover a broader spectrum of risks associated with LLMs. For example, even with a very low rate of toxicity in BOLD, we observe strong biases against some demographics in that benchmark.
> - **Intersectional Demographic Analysis:** Examining the intersection of demographics, we observed instances where toxic generations are more frequent than with  individual demographics. This underscores the importance of exploring intersectional demographics in future benchmarking efforts.
> - **Mitigation Techniques and Their Effects:** Smaller LLMs exhibited more effectiveness with self-debiasing, while prompting was more successful with larger models like BlenderBot3-175B. We hypothesize that the much larger capacity of BlenderBot3-175B may make it much more capable of adjusting its output via prompting. We also note a general reduction in biases as per human evaluations.
> - **The Influence of Training data on Bias:**  Our investigation into distribution of demographic terms in common training corpora identified the underrepresentation of gender non-conforming and sex minority terms, potentially contributing to systematic biases against LGBTQ+ minority groups.
>
> Based on the above key takeaways, we revised the **Conclusion section** as presented below:
> ```
> 	In our analysis, we find that each prompt dataset causes the LLM models to output generations with different rates of toxicity and negative regard. Notably, even when the baseline toxicity rate is minimal,certain demographic biases manifest prominently across specific prompt datasets. Moreover, the prompt datasets studied in this paper, when used in combination with each other, are able to surface a more diverse set of risks posed by LLMs, providing a holistic view into which subgroups may be at higher risk of marginalization by LLMs. We hope that our measurement results show how multi-metric measurement can enable us to better understand the possible risks LLMs can pose, and can better expose at-risk groups that may be affected. We accentuate the significance of assessing toxicity and bias concerning intersectional demographics, underscoring instances where toxic content frequency surges for these groups in contrast to individual demographics.
> Moreover, we explored several mitigation techniques gauging their efficacy via both automated metrics and human evaluation. We observed that self-debiasing technique is mostly effective in smaller LLMs, while prompting is more effective in larger LLMs. We hypothesize that the much larger capacity of larger LLMs may make them much more capable of adjusting their output via prompting. Moreover, these techniques exhibit  promising impact in mitigating biases, a finding that encourages further research into their enhancement and expansion for pre-trained LLMs, in addition to instruction-tuning, and RLHF, which apply at later stages of model training.
>
> Analyzing the demographic distribution in common training corpora, we unveiled an underrepresentation of gender non-conforming and sex minority terms. This potentially enhances biases against LGBTQ+ minority groups in LLMs.
>
> We aspire for LLMs to effortlessly generate respectful and insightful content about all demographics. Using diverse datasets together helps us analyze bias in a more inclusive way. While the list of demographics and subgroup labels in each prompt dataset is not fully comprehensive, ongoing expansion will boost the inclusiveness of bias analysis. This list of relevant subgroups should evolve constantly to reflect societal and cultural changes. In light of our findings, we recognize the tendency for toxicity and negative regard to escalate with model size. Given the rapid development of larger LLMs and the widespread use of RLHF models, future endeavors could concentrate on establishing benchmarks to assess bias and toxicity within instruction-tuned models.
> Moving forward, we envision the field's progression towards improved and widespread utilization of multi-metric bias measurements similar to our exemplified approach, enabling a more comprehensive evaluation of models across a broad spectrum of potential biases.
> ```
>
> > I can not assess the practical value of this paper to follow up work given that the source code of the analyses is not referenced, nor is it available in supplementary materials. Regretfully, the sole promise of publishing it after the peer review does not allow for the evaluation of its practical usability.
>
> Once the condition of anonymity is lifted, we plan to publish a Github repository containing code needed to reproduce the measurements reported in our paper. This includes logic for loading the benchmark datasets, an extensible interface for running inference on the models, and routines for calculating the final metrics including the BiasScore. Our work is based on many open source libraries that are widely used in the community (e.g. PyTorch, HuggingFace Transformers, etc.) which we hope increases its usefulness and chance of adoption.
>
> > In Section 3.2, you also confronted the human evaluation results, but I could not find any quantified reports regarding this. Are the details of the reported human evaluation available somewhere?
>
> In response to your **question**, we thank you for pointing out the lack of clarity, we included human evaluation methods and results in Section C.2.4, and we’ve added a statement in Section 3.2 to check Section C.2.4 for more details on human evaluations.
>
> > Could not reproduce the results here no matter how hard they tried.
>
> Regarding you concern about **reproducibility**:  In the paper, we have detailed our model decoding parameter settings in B.2 and settings of different bias/toxicity mitigation techniques in B.3 to enable the reproducibility of our paper. Apart from the two proposed metrics (AdvPromptSet and HolisticBiasR), the other four metrics that are compared in the paper are public research benchmarks. Furthermore, as mentioned in the paper, we are committed to open sourcing the proposed two new datasets (AdvPromptSet and HolisticBiasR) as well as the evaluation pipeline and bias and toxicity mitigation toolkit for robust measurement across different metrics, fostering reproducibility and benefiting the broader research community. This initiative reinforces the reliability of our paper's findings.
>
> Thank you again for your insightful feedback, we hope we were able to address your concerns. Please let us know if there are any additional questions or concerns.

---

### Official Review · Reviewer_SybY · 2023-08-05

**Soundness:** 4

**Excitement:**

3: Ambivalent: It has merits (e.g., it reports state-of-the-art results, the idea is nice), but there are key weaknesses (e.g., it describes incremental work), and it can significantly benefit from another round of revision. However, I won't object to accepting it if my co-reviewers champion it.

**Paper Topic And Main Contributions:**

The paper focuses on robust bias evaluation of large language models. They analyze 5 model families of LLMs across 12 demographic axes and 6 bias and toxicity datasets. In addition to existing datasets, they construct their AdvPromptSet that evaluate on specific interactions of gender, race, etc. They also compare mitigation techniques with their evaluations.

**Reasons To Accept:**

1. Robust bias evaluation is an important research direction.
2. The paper performs a systematic analysis of a wide range of LLMs with a focus on biases.
3. Their constructed dataset can be helpful for future work.

**Reasons To Reject:**

1. It is hard to draw insightful conclusions from their experiments. While the experiments are relatively comprehensive, it is hard to draw insights or provide guidelines for future research in this direction.
2. Related to the first point, the paper tries to include many things, such as constructing a dataset, evaluating various LLMs and mitigation strategies, etc. However, each of the sections is not discussed in detail (e.g. the details of AdvPromptSet) and it would be better to just focus on one or two aspects and better emphasize their contributions and findings.
3. The paper should be better organized and make the connections between each part more clear.

**Reproducibility:**

3: Could reproduce the results with some difficulty. The settings of parameters are underspecified or subjectively determined; the training/evaluation data are not widely available.

**Reviewer Confidence:**

3: Pretty sure, but there's a chance I missed something. Although I have a good feel for this area in general, I did not carefully check the paper's details, e.g., the math, experimental design, or novelty.

---

> ### Author Rebuttal · Authors · 2023-08-29
>
> Thank you for your thoughtful assessment of our paper! Your recognition of the importance of robust bias evaluation as a research direction, along with the acknowledgement of our systemic analysis of various LLMs and the utility of our novel datasets for future research, is truly encouraging.
>
> Below please find our honest effort to address your concerns.
>
> > It is hard to draw insightful conclusions from their experiments. While the experiments are relatively comprehensive, it is hard to draw insights or provide guidelines for future research in this direction.
>
> Thank you for highlighting the need for greater clarity in conveying our main findings and for suggesting a restructuring of our conclusion sections to address this concern. We intend to address these issues in our revised manuscript by presenting the primary insights derived from our experiments, which will be highlighted in the following manner:
> - **Impact of Model Size on Bias and Toxicity:** Toxicity and negative regard often increases as a function of model size, but not always. Our findings align with previous research, and highlights the importance of conducting robust bias analysis on modern LLMs.
> - **Consistent bias towards specific Demographics:** Our analysis revealed a consistent negativity towards some demographics such as Hispanic or Latino across all LLMs. Considering the growing integration of LLMs into our daily lives, this bias could have significant implications for these minority groups.
> - **Differential Effects of Benchmark Datasets:** Each prompt dataset led to varying levels of BiasScore, toxicity and negative regard in the generated content of LLMs. The substantial bias indicated by our BiasScore results across all LLMs highlights the necessity of employing diverse benchmark datasets to uncover a broader spectrum of risks associated with LLMs. For example, even with a very low rate of toxicity in BOLD, we observe strong biases against some demographics in that benchmark.
> - **Intersectional Demographic Analysis:** Examining the intersection of demographics, we observed instances where toxic generations are more frequent than with  individual demographics. This underscores the importance of exploring intersectional demographics in future benchmarking efforts.
> - **Mitigation Techniques and Their Effects:** Smaller LLMs exhibited more effectiveness with self-debiasing, while prompting was more successful with larger models like BlenderBot3-175B. We hypothesize that the much larger capacity of BlenderBot3-175B may make it much more capable of adjusting its output via prompting. We also note a general reduction in biases as per human evaluations.
> - **The Influence of Training data on Bias:**  Our investigation into distribution of demographic terms in common training corpora identified the underrepresentation of gender non-conforming and sex minority terms, potentially contributing to systematic biases against LGBTQ+ minority groups.
> - In addition, we have outlined potential **future research directions** based on our key findings:
>     - **Evolving Benchmarks which contain Intersectional Demographics:** Given the gaps in existing benchmarks and the significance of intersectional demographics analysis, a constantly evolving benchmark reflecting societal changes is imperative.
>     - **Benchmarks for Instruction-Tuned Models:** With the proliferation of larger LLMs and reinforcement learning from human feedback (RLHF) models, there is a need for benchmarks that cater to instruction-tuned models, given toxicity and negative regard frequency often increases as a function of model size.
>     - **Targeted Mitigations for Specific Groups:** The higher negativity observed towards certain groups across all LLMs highlights the requirement for tailored mitigations to address toxicity for marginalized communities facing more frequent toxic content.
>     - **Applying Mitigations prior to RLHF and Instruction Fine-tuning:** Given the effectiveness of certain mitigation approaches in pre-trained LLMs, future research could explore their application to mitigate toxicity and bias in their instruction-tuned variants as an additional step before or after RLHF and instruction fine-tuning.
>
> > Related to the first point, the paper tries to include many things, such as constructing a dataset, evaluating various LLMs and mitigation strategies, etc. However, each of the sections is not discussed in detail (e.g. the details of AdvPromptSet) and it would be better to just focus on one or two aspects and better emphasize their contributions and findings.
>
> We thank the reviewer for helping us to improve the readability of the paper. We agree that the discussion about the AdvPromptSet creation should be moved to the main paper. As stipulated by the submission guidelines, we are currently prohibited from making any revisions to the paper prior to receiving the acceptance or rejection notification. However, if our paper is accepted, we are committed to implementing the proposed changes. Specifically, we will transfer the comprehensive information about the dataset creation, detailed in section B.1.2, along with Figure 2, to the main body of the paper.
>
> > The paper should be better organized and make the connections between each part more clear.
>
> If the paper is accepted, we will reorganize the paper for a clearer presentation. Our focus is two folds: 1) **Benchmarking:** a comparison of 6 different prompt-based bias and toxicity metrics across 12 demographic axes and 5 families of generative LLMs, out of those 6 metrics two (AdvPromptSet and HolisticBiasR) of them are novel datasets proposed in the paper. 2) **Mitigation:** the comparison of those benchmarks gives us insights about the bias and toxicity of the compared models. Therefore, we explore the frequency of demographic terms in LLMs common pre-training corpora and how this may relate to model biases. Furthermore, we conduct a comprehensive study of how well 3 mitigation techniques perform across our suite of measurements. Our paper aims to provide insights for partitioners: when deploying a model, they not only have to measure potential harms, but also to understand how they arise (including by characterizing the data) and mitigate them once found, balancing any tradeoffs.
>
> > Could reproduce the results with some difficulty. The settings of parameters are underspecified or subjectively determined; the training/evaluation data are not widely available.
>
> Regarding you concern about **reproducibility**, we would like to iterate that we are committed to open sourcing the proposed two new datasets (AdvPromptSet and HolisticBiasR) as well as the evaluation pipeline and bias and toxicity mitigation toolkit for robust measurement across different metrics, fostering reproducibility and benefiting the broader research community. This initiative reinforces the reliability of our paper's findings. The other four metrics that are compared in the paper are public research benchmarks. We have also detailed our decoding parameter settings in B.2 and settings of different bias/toxicity mitigation techniques in B.3 to enable the reproducibility of our paper.
>
> Your insights and feedback provide valuable validation for our work and improve the presentation of our paper. We are grateful for your thoughtful review. Please let us know if you have any questions or concerns.

---

### Meta-Review · Area_Chair_EDqB · 2023-09-18

**Recommendation:** 4

**Metareview:**

In their long paper submission, “ROBBIE: Robust Bias Evaluation of Large Generative Language Models“, the authors propose a bias/ toxicity assessment benchmark consisting of 12 demographic axes and 6  bias and toxicity datasets (including 2 novel ones) using which they test 5 model families and 3 bias mitigation techniques.

The reviewers value the rich set of the experiments with high soundness scores, and also the transparent discussion of the limitations of this work. The main claims of this work are supported! (At the same time, the authors did not release the code (yet) which hinders reproducibility.)
Compared to the soundness assessment of this work, the excitement of the reviewers is a bit more limited. This is, for instance, due to the practical impact being rather unclear.

---

### Decision · Program_Chairs · 2023-10-07

**Decision:**

Accept-Main

**Comment:**

In their long paper submission, “ROBBIE: Robust Bias Evaluation of Large Generative Language Models“, the authors propose a bias/ toxicity assessment benchmark consisting of 12 demographic axes and 6  bias and toxicity datasets (including 2 novel ones) using which they test 5 model families and 3 bias mitigation techniques.

The reviewers value the rich set of the experiments with high soundness scores, and also the transparent discussion of the limitations of this work. The main claims of this work are supported! (At the same time, the authors did not release the code (yet) which hinders reproducibility.)
Compared to the soundness assessment of this work, the excitement of the reviewers is a bit more limited. This is, for instance, due to the practical impact being rather unclear.